# Design and Optimization of an S-Band MEMS Bandpass Filter Based on Aggressive Space Mapping

**DOI:** 10.3390/mi14010067

**Published:** 2022-12-27

**Authors:** Qiannan Wu, Xudong Gao, Zemin Shi, Jing Li, Mengwei Li

**Affiliations:** 1School of Semiconductors and Physics, North University of China, Taiyuan 030051, China; 2Academy for Advanced Interdisciplinary Research, North University of China, Taiyuan 030051, China; 3Center for Microsystem Integration, North University of China, Taiyuan 030051, China; 4School of Instrument and Intelligent Future Technology, North University of China, Taiyuan 030051, China; 5Key Laboratory of Dynamic Measurement Technology, North University of China, Taiyuan 030051, China; 6School of Instrument and Electronics, North University of China, Taiyuan 030051, China

**Keywords:** ASM, interdigital filter, double-layer, SIR, MEMS

## Abstract

Aggressive space mapping (ASM) is a common filter simulation and debugging method. It plays an important role in the field of microwave device design. This paper introduces ASM and presents the design and fabrication of a compact fifth-order microstrip interdigital filter with a center frequency of 2.5 GHz and a relative bandwidth of 10% using ASM. The filter used a double-layer silicon substrate structure and stepped impedance resonators (SIRs) and was optimized by ASM. After five iterations, the filter achieved the design specification, which greatly improves the efficiency of the filter design compared with the traditional method. It was fabricated on high-resistance silicon wafers by micro-electro-mechanical systems (MEMSs) technology, and the final size of the chip is 9.5 mm × 7.6 mm × 0.8 mm. The measurement results show that the characteristics of the filter are similar to the simulation results, which also shows the efficiency and precision of the ASM algorithm.

## 1. Introduction

Filters can select or suppress RF signals in a specific frequency range, thus combining or separating signals of different frequencies [1,2]. With the booming development of 5G communication technology [3], various microwave and millimeter wave communication systems and devices are widely used in the fields of satellite, radar, aviation, marine communication, and mobile communication [4]. Microwave filters, as an important part of them, play an irreplaceable role in frequency selection, suppressing various kinds of clutter and reducing noise interference [5,6]. Better performance, smaller size, lighter weight, and lower cost microwave filters have become the most urgent needs. Therefore, how to quickly and efficiently develop a high-performance and miniaturized filter is currently a key aspect of microwave RF filter design. The auxiliary diagnosis and debugging of filters have attracted more and more attention [7].

Space mapping [8] is an efficient means of debugging and optimizing for microwave and millimeter wave devices. It was first proposed in 1994 by Bandler and called original space mapping (OSM). It can establish a linear mapping relationship between two models (coarse model and fine model) through a large number of samples. Subsequently, an improvement in aggressive space mapping (ASM) was proposed to simplify the complex process and solve some non-linear issues [9]. Aggressive space mapping can simply establish initial mapping relations. Then, the mapping relationship between the two spaces is updated by parameter extraction. It is widely applicable to various nonlinear mapping problems [10]. Zhang et al. [11] designed a silicon-based bandpass filter with a center frequency of 24 GHz exploiting ASM. After only four iterations, the filter achieved the design specification, but the problem of high losses in the single-layer silicon substrate was also exposed. Xiao et al. [12] designed a double-layer silicon-based stripline filter with a center frequency of 20.76 GHz. After five iterations, the filter meets the design requirements. This filter had a lower loss, but its size is too big for a microwave system. In addition, many researchers used ASM in the debugging of filters and verified the feasibility and effectiveness of ASM in filter design [13,14,15,16,17].

To overcome the problems of large size and high loss of silicon-based filters, this paper proposes a filter structure with a double-layer silicon substrate and stepped impedance resonators (SIRs) [18,19,20]. As shown in Figure 1, this structure is similar to substrate integrated waveguide (SIW) [21], the double-layer high resistance silicon substrate, and surrounding TSV metalized vias enclosing the transmission line. The coplanar waveguide with a characteristic impedance of 50 Ω was used for the input and output interface of the filter. The structure not only reduces electromagnetic leakage and improves filter performance, it also reduces the filter volume.

## 2. Theory and Design

### 2.1. General Formulation of ASM

Aggressive space mapping is an efficient optimization algorithm that is widely used in the design of microwave devices and microwave circuits. It establishes a dynamic mapping relationship between the coarse model $x_{c}$ and fine model $x_{f}$ so that the complex and time-consuming optimization problem of the fine model is transformed into simple and fast optimization and updating of the coarse model. While realizing fast and efficient optimization, some nonlinear problems can be resolved.

In the process of solving by the ASM algorithm, $x_{c}^{\left(*\right)}$ represents the optimal design parameter of the coarse model and $P\left(x_{f}\right)$ represents the coarse model response. The coarse model and fine model meet the following requirements in a small range:(1)$$P\left(x_{f}\right)-x_{c}^{\left(*\right)}=0$$

The next design parameters of the fine model are updated by the quasi-Newton iteration, as follows:(2)$$x_{f}^{\left(j+1\right)}=x_{f}^{\left(j\right)}+h^{\left(j\right)}$$
where $h^{\left(j\right)}$ is given by
(3)$$\left\{\begin{array}{l} h^{\left(j\right)}=-{\left(B^{\left(j\right)}\right)}^{-1}f^{\left(j\right)}\\ f^{\left(j\right)}=P\left(x_{f}^{\left(j\right)}\right)-x_{c}^{\left(*\right)}=x_{c}^{\left(j\right)}-x_{c}^{\left(*\right)} \end{array}\right.$$

If the iteration step length is a quasi-Newtonian step length, the solution of B can be updated using the Broyden formula, which satisfies the quasi-Newton update formula:(4)$$B^{\left(j+1\right)}=B^{\left(j\right)}+\frac{f^{\left(j+1\right)}}{h^{\left(j\right)}{}^{T}h^{\left(j\right)}}h^{\left(j\right)}{}^{T}$$

When the space physical meaning of the coarse model and fine model is the same,
(5)$$B^{\left(0\right)}=I$$
where ***I*** is the identity matrix and the super-index ***T*** stands for transpose.

### 2.2. Basic Principles of SIR

Uniform impedance resonators (UIRs) [22] and step impedance resonators (SIRs) are common transmission line type resonators, widely used in microwave filters design. Step impedance resonators (SIRs) are simple in structure, easy to design, and able to reduce the length of the resonator without changing the unloaded Q value. Compared with UIR, SIR has more advantages in terms of structural form, harmonic suppression, design parameter control, and filter miniaturization.

Stepped impedance resonators (SIRs) are a kind of resonator composed of more than two transmission lines with different characteristic impedance and capable of transmitting TEM mode or quasi-TEM mode. It comes in three types: $\lambda_{g}/4$, $\lambda_{g}/2$, and $\lambda_{g}$, where the $\lambda_{g}/4$-type resonators, owing to a small volume and the fact that the parasitic frequency band is three times the center frequency, have been widely used in small volume filters’ design. As shown in Figure 2, the characteristic impedance and equivalent electrical length of the open-circuit end and short-circuit end are (*Z*_1_,*θ*_1_) and (*Z*_2_,*θ*_2_), respectively.

According to Figure 2, its input impedance can be expressed as follows:(6)$$Z_{in}=jZ_{2}\frac{Z_{1}\tan\theta_{1}+Z_{2}\tan\theta_{2}}{Z_{2}-Z_{1}\tan\theta_{1}\tan\theta_{2}}$$

Defining the impedance ratio $R_{Z}=Z_{2}/Z_{1}$, when *Y*_in_ = 1/*Z*_in_ = 0, as in Equation (7), the $\lambda_{g}/4$-type SIR achieves the resonance condition.
(7)$$R_{Z}=Z_{2}/Z_{1}=\tan\theta_{1}\tan\theta_{2}$$

Therefore, the $\lambda_{g}/4$-type resonance conditions are related to $\theta_{1}$, $\theta_{2}$, and $R_{Z}$. Compared with ordinary microstrip line resonators, $\lambda_{g}/4$-type SIR has one more design degree of freedom, which has greatly promoted the specific application of various SIR resonators.

In general, compared with conventional uniform impedance resonators (UIR) whose electric length is π/2, the normalized resonator length Ln of $\lambda_{g}/4$-type resonators is as follows:(8)$$L_{n}=\theta_{TA}/\left(\pi /2\right)=2\theta_{TA}/\pi$$

Figure 3 is the curve graph between the normalized resonator length Ln and the electric length $\theta_{1}$ when the impedance ratio R_Z_ is different. According to Figure 3, in the process of designing filters using SIR, a smaller impedance ratio $R_{Z}$ should be chosen so that the filter size can be reduced and the original transmission performance of the resonator can be maintained as much as possible. In the design process, the commonly used impedance ratio $R_{Z}$ = 0.8 [23].

### 2.3. Design and Optimization

In this paper, a fifth-order microstrip interdigital filter was designed with a center frequency of 2.5 GHz, a bandwidth of 0.25 GHz, and a passband Ripple of 0.1 dB. The 2D structure of this filter is shown in Figure 4.

The ASM algorithm involves coarse and fine models that are based on physical parameters. Then, a mapping relation is established between the two models using a mapping matrix for quick design and optimization. The fine and coarse models of this filter are represented by the HFSS model and its normalized coupling matrix *M*. According to the design index of the filter, the normalized coupling matrix *M* of the filter is obtained by the Chebyshev synthesis method [24]:$$M=\left[\begin{array}{ccccccc} 0 & 0.933 & 0 & 0 & 0 & 0 & 0\\ 0.933 & 0 & 0.797 & 0 & 0 & 0 & 0\\ 0 & 0.797 & 0 & 0.607 & 0 & 0 & 0\\ 0 & 0 & 0.607 & 0 & 0.607 & 0 & 0\\ 0 & 0 & 0 & 0.607 & 0 & 0.797 & 0\\ 0 & 0 & 0 & 0 & 0.797 & 0 & 0.933\\ 0 & 0 & 0 & 0 & 0 & 0.933 & 0 \end{array}\right]$$

Based on the normalized coupling matrix *M*, the coupling coefficient Kij and the external quality factor *Qe* of the filter can be derived. *c* is a vector representing the ideal center frequency, coupling coefficient, and external quality factor:*c* = [*f*_1_ = 2.5 GHz, *f*_2_ = 2.5 GHz, *f*_3_ = 2.5 GHz, m12 = 0.797, m23 = 0.607, *Qe* = 11.48]

Therefore, six physical quantities need to be optimized, and the parameter optimization vector of the coarse model is as follows:*x_c_* = [L1, L2, L3, S1,2, S2,3, Lt]^T^

Then, the resonant frequency *f_0_* and external quality factor *Qe* of the single resonator and the coupling coefficient *K* between two resonators of this filter are extracted by the 3D electromagnetic simulation software Ansoft HFFSS’s eigenmode solution model. The extraction of coupling coefficients *K* requires the creation of two single-cavity resonators placed side-by-side. At this point, the inherent resonant frequency *f_0_* of the resonator is decomposed into two new frequencies *f_1_* and *f_2_* owing to the coupling effect. The coupling coefficient *K* can be solved by *f_1_* and *f_2_* as in Equation (9):(9)$$K=\frac{f_{2}^{2}-f_{1}^{2}}{f_{2}^{2}+f_{1}^{2}},$$

The simulation relationship between these physical quantities (*f*_0_, *Qe*, and *K*) and optimization variables is established, as shown in Figure 5.

According to Figure 5, the initial physical size of the filter x = [385, 457, 1380, 8046, 8046, 8046]T, and its unit is um. Then it was modeled and simulated by Ansoft HFFSS by Ansoft Corporation, Pittsburgh, PA, USA. Its initial response is shown in Figure 6a. This did not meet the index requirements. The optimization process of applying ASM to the bandpass filter is described as follows:(1)Initialize j=1, $B^{\left(1\right)}=I$, extract the S parameter of the initial response by the Cauchy method and calculate the corresponding coupling matrix. Calculate $x_{c}^{\left(1\right)}$ through Figure 5, and obtain the current mapping error $f^{\left(1\right)}=x_{c}^{\left(1\right)}-x_{c}^{\left(*\right)}$;(2)Calculate fine model design parameter growth step $h^{\left(1\right)}$ using Equation (3), and work out the design parameters for the next step of the fine model $x_{f}^{\left(2\right)}=x_{f}^{\left(1\right)}+h^{\left(1\right)}$;(3)Perform the fine mode simulation in HFSS and obtain the responses in Figure 6b, which still does not meet the design index and needs to be further optimized;(4)Perform parameter extraction, extract the design parameters of the rough model $x_{c}^{\left(2\right)}$, and calculate the mapping error $f^{\left(2\right)}$;(5)Update the mapping relation matrix $B^{\left(1\right)}$ to $B^{\left(2\right)}$ through Equation (4);(6)j=j+1, return to step 3 and continue executing the program until the response of the filter satisfies the design metrics.


As shown in Figure 6f, after the fifth iteration, the filter satisfies the design specifications.

The optimized parameters of the filter after each iteration are shown in Table 1.

## 3. Fabrication and Measurement

The above-mentioned interdigital filter can be fabricated by MEMS processing technology, which mainly includes bulk silicon, microsurface, and metallic bonding technology. Its substrate is a kind of high-resistivity silicon (>10 KΩ·cm) with a thickness of 400 µm. The main process steps involved are photolithography, ICP etching, coating, electroplating, corrosion, and bonding. The fabrication process is shown in Figure 7.

Firstly, a 10-micron thick photoresist was spin-coated on the silicon wafer surface and the pattern of the holes was transferred to the photoresist by photolithography. Secondly, holes were etched on the silicon wafer by the inductively coupled plasma (ICP) etching process. Thirdly, a 0.5-micron thick SiO_2_ isolation layer was then grown on the surface of the wafer and the sidewalls of the TSV by a high-temperature thermal oxygen process. Fourthly sputtered adhesion and seed layers of 50 nm Ti and 250 nm Au, respectively, on the wafer surface; then electroplate resonator and ground plane on wafer surface using photoresist as plating die and removing the seed layer and adhesion layer by wet etching. Finally, Au–Au thermal-compression bonding. The final size of the chip is 9.5 mm × 7.6 mm × 0.8 mm. The photograph of the fabricated filter is shown in Figure 8.

The filter was tested with a PNA-N5227B vector network analyzer manufactured by KEYSIGHT (Santa Rosa, CA, USA) with a measurement range of 10 MHz to 67 GHz. Furthermore, a pair of TITAN-RC-200 probes and a YB600 probe station were used. A comparison of the simulation and measurement results is shown in Figure 9. It can be seen that the simulation curve and the actual measured curve do not coincide completely, but they are similar. The simulation results show that the insertion loss of this filter at 2.5 GHz is 2.1 dB, the out-of-band rejection is 40 dB, and the return loss is 15 dB; the measured results show that the insertion loss of this filter at 2.5 GHz is 3.2 dB, the out-of-band rejection is 21 dB, and the return loss is 17 dB. By considering the errors in processing and measurement, the measured performance of the filter is reduced, but within a certain range, the method of designing and optimizing the filter using ASM can be adopted.

## 4. Conclusions

This paper mainly introduces the application prospect of the bandpass filter and the important role of ASM in microwave device design. Then, a compacted S-band interdigital bandpass filter with stepped impedance resonators (SIRs) was designed with ASM after five iterations. It was fabricated on high-resistance silicon wafers by the MEMS process, which is conducive to the miniaturization and integration of communication systems. The test results and design results of the filter remain highly consistent, despite some differences caused by process errors. In summary, aggressive space mapping is a fast and efficient method to optimize the design of microwave devices.

## Figures and Tables

**Figure 1 micromachines-14-00067-f001:**
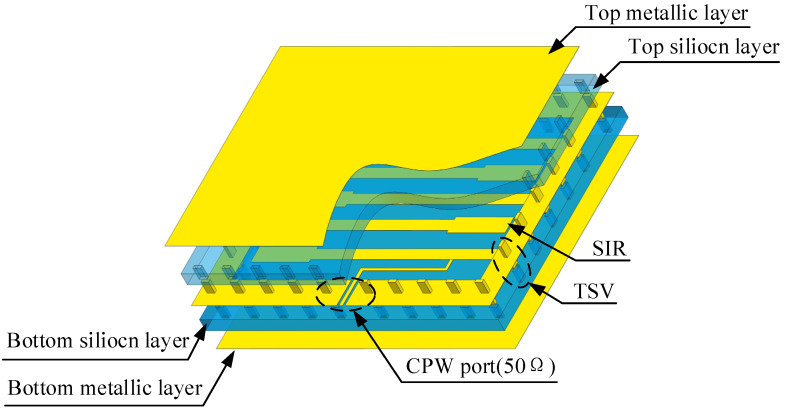
The structure of the double-layer silicon-based interdigital filter.

**Figure 2 micromachines-14-00067-f002:**
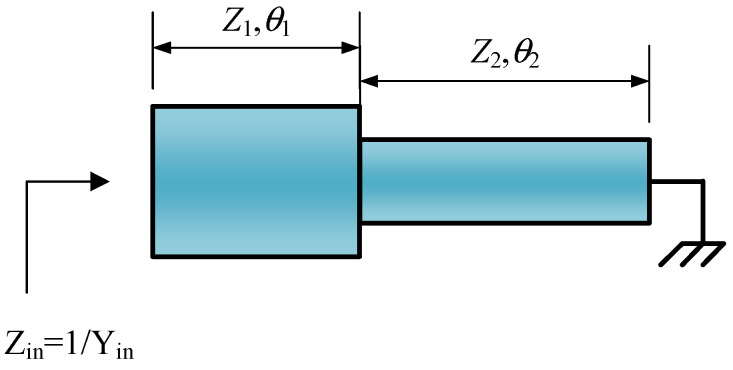
The basic structure of a λg/4-type SIR.

**Figure 3 micromachines-14-00067-f003:**
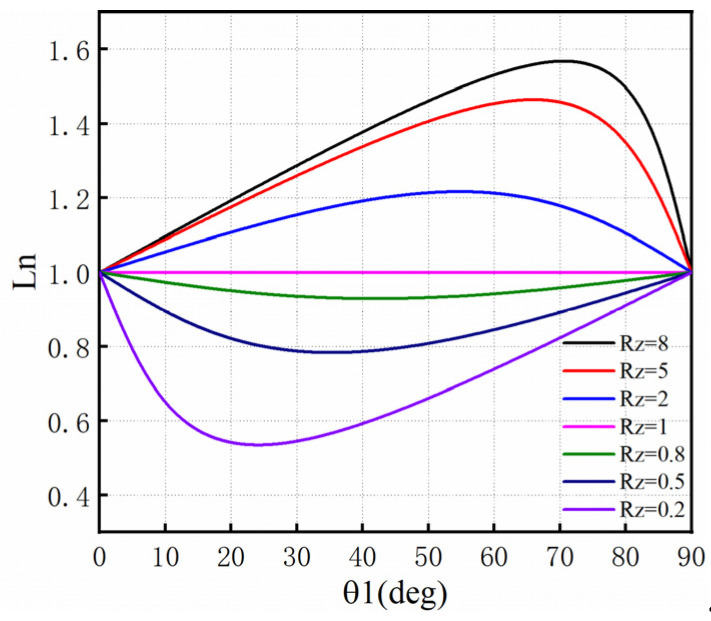
Resonance conditions of SIR.

**Figure 4 micromachines-14-00067-f004:**
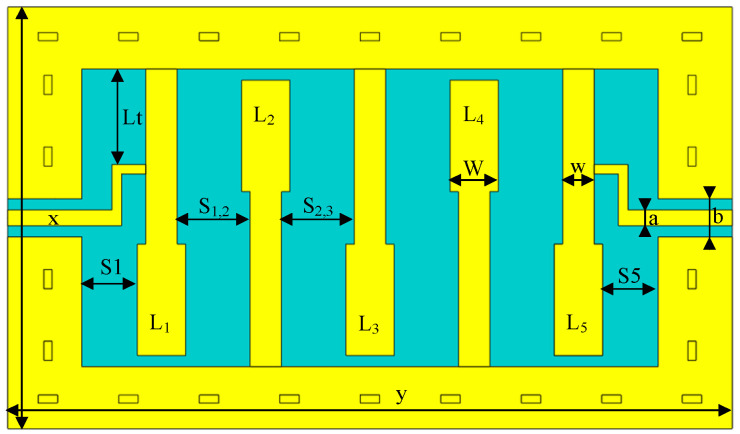
Two-dimensional structure of the fifth-order interdigital filter.

**Figure 5 micromachines-14-00067-f005:**
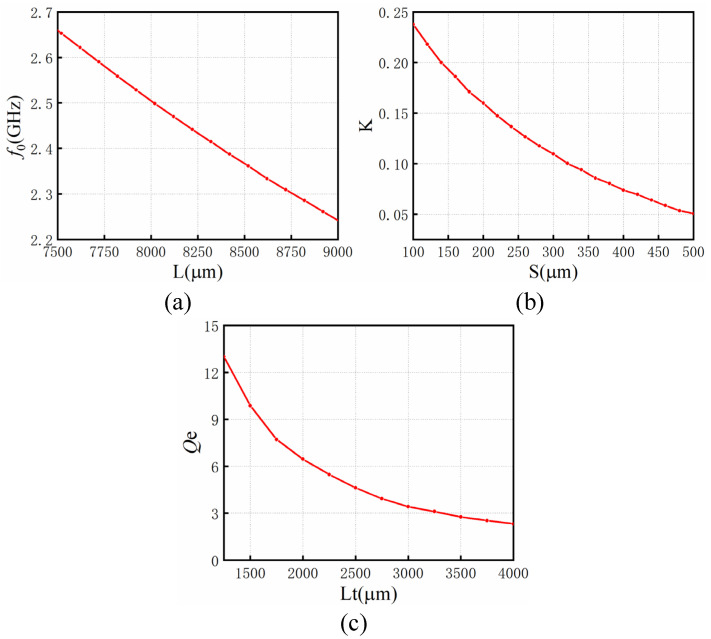
(**a**) Relationship between center frequency *f*_0_ and resonator length L. (**b**) Relationship between coupling coefficient K and distance S. (**c**) Relationship between external quality factor *Qe* and tapped position Lt.

**Figure 6 micromachines-14-00067-f006:**
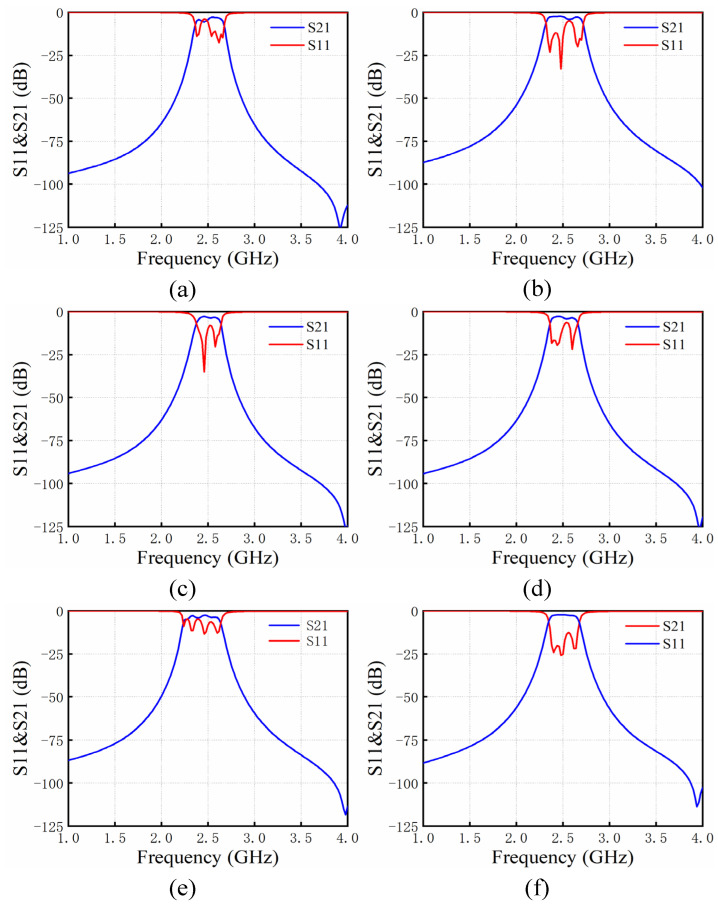
(**a**) Initial response of the filter. (**b**) Response after the first iteration. (**c**) Response after the second iteration. (**d**) Response after the fourth iteration. (**e**) Response after the fifth iteration. (**f**) Response after the sixth iteration.

**Figure 7 micromachines-14-00067-f007:**
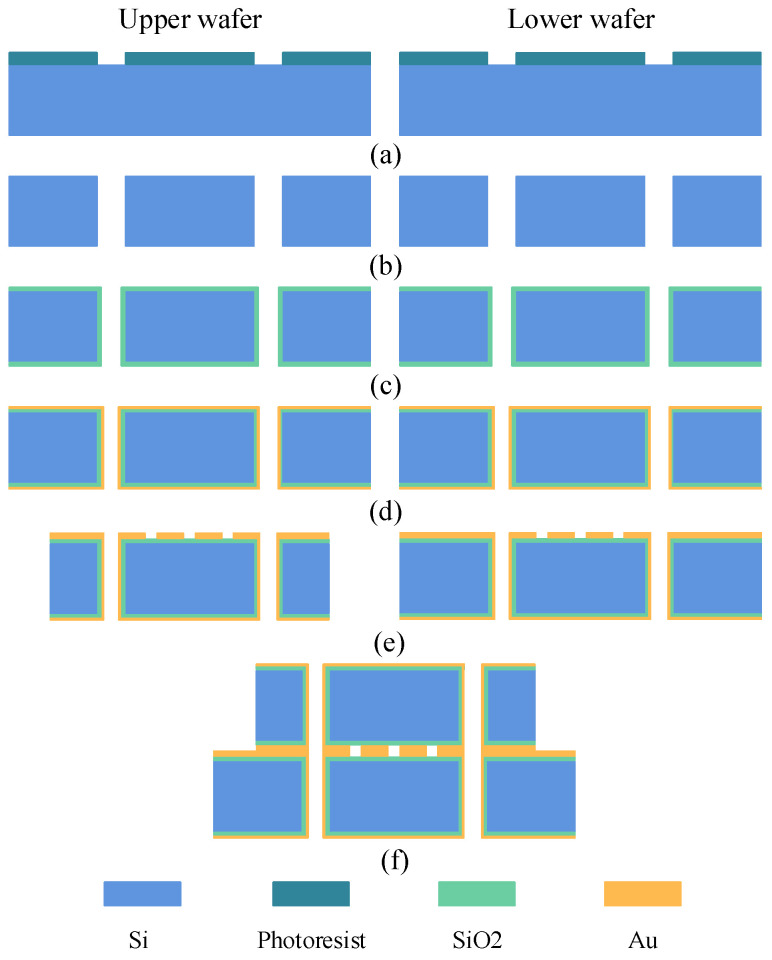
The fabrication processes of the interdigital filter. (**a**) Pattern the hole photoresist. (**b**) Etching via-hole. (**c**) Preparation of isolation layer. (**d**) Sputter the seed layer. (**e**) Electroplating surface structure. (**f**) Au–Au bonding.

**Figure 8 micromachines-14-00067-f008:**
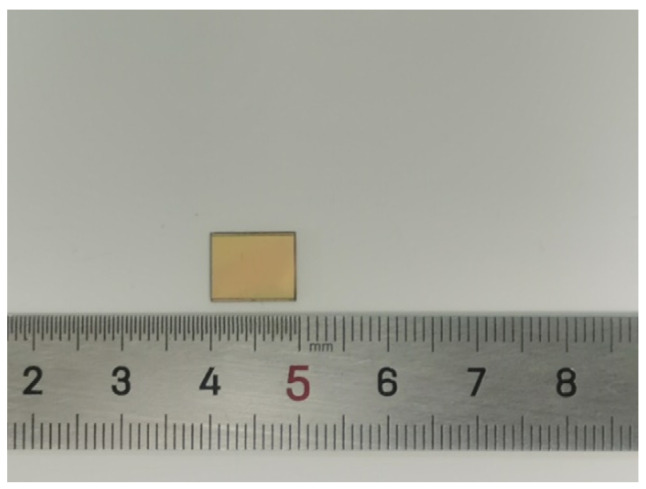
Photograph of the fabricated filter.

**Figure 9 micromachines-14-00067-f009:**
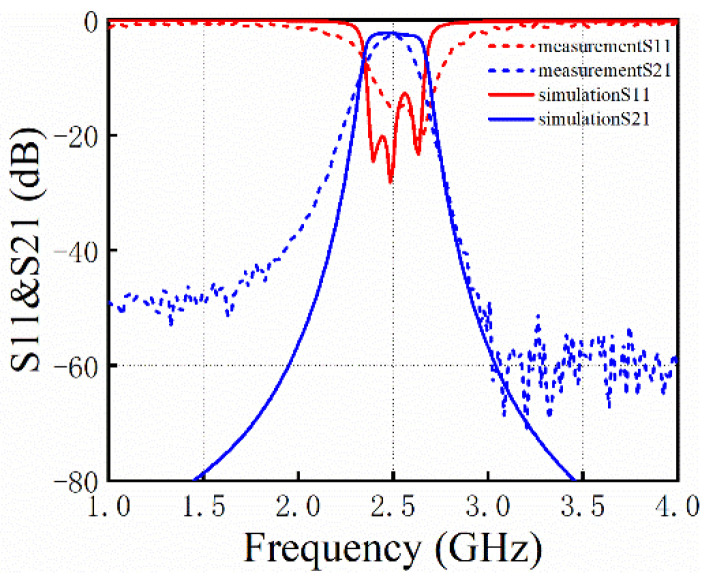
Simulation and measurement results of the proposed filter.

**Table 1 micromachines-14-00067-t001:** The optimization parameter in each iteration (unit: µm).

No. of Iterations	L_1_	L_2_	L_3_	S_1,2_	S_2,3_	L_t_
Initial value	8046	8046	8046	385	457	1380
1st iteration	8082	7703	7703	386	458	1367
2nd iteration	8124	7745	7682	459	540	1402
3rd iteration	8034	7741	7689	467	520	1266
4th iteration	8108	7610	7759	431	515	1336
5th iteration	7720	7680	7680	407	507	1520

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
