# Peer review of "Design and Optimization of an S-Band MEMS Bandpass Filter Based on Aggressive Space Mapping"

_micromachines, 2022, doi:10.3390/mi14010067_

Round 1

Reviewer 1 Report

The manuscript presents a S-band bandpass filter designed on silicon substrate with a space mapping optimization technique. 

The general information of ASM is informative to readers, but, in the next section of analyzing stepped impedance resonator, the authors need to clearly claify the reason why the SIR is suitable or selected to the proposed filter desingn. Usually, the SIR is used for the stopband improvement, so in the measurement section, the measured stopband response should be included if the author intended to improve the stopband performance by using SIR.

The discrepancy between results of simulation and measurement is too much to ignore. The author needs to mention the reason of the difference at the passband and stopband in detail.

Author Response

Dear Reviewers1:
 I have uploaded the response information in the attachment and thank you very much for the advice you gave.

Reviewer 2 Report

Dear authors,

I really appreciate this paper, I find the method really interesting. However, I  have some questions that needs to be answered. I hope you can read my comments directly in the attached pdf-file and make improvements.

As a general comment I think you need a space before [ref], I put some in yellow at the introduction, but there are others. The legends in the figures are really tiny, please make readable.

Also, given the refs [18]-[20] I don't really see what is the new contribution of your work, please argument the added value.

Finally, you claim that the difference between simulated and measured results are due to processing errors. Can you then specify what errors are made and do a simulation according to these errors to see if it holds up.

Thank you

Author Response

Dear Reviewers2.
 I have uploaded the response information in the attachment and thank you very much for the advice you gave.

Round 2

Reviewer 1 Report

The authors reply well to all previous concerns.

The insertion and return losses of filters should be given with positive numbers, so the reviewer recommends that the authors not add the minus signs in the measurement section.

Author Response

Dear Reviewers1:
I have uploaded the reply information, thank you very much for your advice.
